# Modeling Overdispersed Dengue Data via Poisson Inverse Gaussian Regression Model: A Case Study in the City of Campo Grande, MS, Brazil

**DOI:** 10.3390/e24091256

**Published:** 2022-09-07

**Authors:** Erlandson Ferreira Saraiva, Valdemiro Piedade Vigas, Mariana Villela Flesch, Mark Gannon, Carlos Alberto de Bragança Pereira

**Affiliations:** 1Institute of Matematics, Federal University of Mato Grosso do Sul, Campo Grande 79070-900, MS, Brazil; 2Faculty of Engineering, Architecture and Urbanism and Geography, Federal University of Mato Grosso do Sul, Campo Grande 79070-900, MS, Brazil; 3Institute of Matematics and Statistics, University of São Paulo, São Paulo 05508-090, SP, Brazil

**Keywords:** dengue fever, poisson regression model, negative binomial regression model, Poisson inverse Gaussian regression model, maximum likelihood estimation

## Abstract

Dengue fever is a tropical disease transmitted mainly by the female *Aedes aegypti* mosquito that affects millions of people every year. As there is still no safe and effective vaccine, currently the best way to prevent the disease is to control the proliferation of the transmitting mosquito. Since the proliferation and life cycle of the mosquito depend on environmental variables such as temperature and water availability, among others, statistical models are needed to understand the existing relationships between environmental variables and the recorded number of dengue cases and predict the number of cases for some future time interval. This prediction is of paramount importance for the establishment of control policies. In general, dengue-fever datasets contain the number of cases recorded periodically (in days, weeks, months or years). Since many dengue-fever datasets tend to be of the overdispersed, long-tail type, some common models like the Poisson regression model or negative binomial regression model are not adequate to model it. For this reason, in this paper we propose modeling a dengue-fever dataset by using a Poisson-inverse-Gaussian regression model. The main advantage of this model is that it adequately models overdispersed long-tailed data because it has a wider skewness range than the negative binomial distribution. We illustrate the application of this model in a real dataset and compare its performance to that of a negative binomial regression model.

## 1. Introduction

According to the world health organization (WHO), dengue fever is a mosquito-borne viral infection that is common in warm, tropical climates. The female of the *Aedes aegypti* mosquito is the main transmitter of the disease, which is caused by four serotypes of a flavivirus, called DENV1, DENV2, DENV3 and DENV4, classified on biological and immunological criteria. As there is still no safe and effective vaccine, the most effective ways to prevent outbreaks of the disease are still to avoid mosquito bites and control the mosquito population [1].

Since the proliferation of the mosquito that transmits dengue depends on temperature, water availability, and some other climatic factors to complete its cycle life, it is of interest to understand the relationships between climatic variables and the recorded number of dengue cases. The Poisson regression (PR) model has been used repeatedly for such applications. For example, Leslie [2] studies the climatic factors that affect the spread of dengue in the city of Colombo, Sri Lanka from the period of 2010 to 2018, using as a primary model a Poisson regression model. Sinaga and Sinulingga [3] model the number of dengue hemorrhagic fever cases in the city of Medan using a Poisson regression model. The authors consider as explanatory variables population density, number of health workers, number of health facilities, area height, and average waste production. Mukhaiyar et al. [4] propose to predict the number of dengue fever cases in Bandung, West Java, Indonesia, in the period 2001–2016, by fitting a Poisson regression model using the temperature and cumulative rainfall as explanatory variables. In these approaches, the observed values for a response variable are taken as having been generated from a Poisson distribution. Using the theory of generalized linear models [5], a log-linear relationship is constructed relating the average value of the response variable to a set of *p* explanatory variables.

An essential assumption of the Poisson regression model is that the mean of the response variable is equal to the variance, a property known as equidispersion. However, dengue-fever data, in general, do not have this property. Therefore, the Poisson regression model is not suitable for modeling such data, because the standard errors may be underestimated, leading to misleading inference from the regression.

For the scenario of overdispersed data, i.e., the variance of the response variable being greater than its average, the usual statistical approach consists of considering a negative binomial regression (NBR) model. Under this approach, it is assumed that the response variable values are generated according to a negative binomial distribution. This distribution is a mixture of a Poisson distribution and a Gamma distribution. Analogously to the PR model, this approach also links the response variable’s average value to a set of *p* explanatory variables by using a log-linear relationship. However, the NBR model is not adequate to model long-tailed datasets, i.e., datasets in which there are some very large integer values far away from the majority [6,7,8].

Therefore, we propose modeling a dengue-fever dataset by using the Poisson-inverse-Gaussian regression (PIGR) model as a competitor to the NBR model. In this model, response-variable values are assumed to be generated according to a Poisson-inverse-Gaussian distribution. This distribution is a mixture of a Poisson distribution and an inverse-Gaussian distribution. The main advantage of this distribution is that it may properly model overdispersed long-tail data because it has a larger range of skewness than a negative binomial distribution [9,10,11,12]. For this model, we also link the expected value of the response variable to a set of *p* explanatory variables by using a log-linear relationship.

We illustrate the fitting of the NBR and PIGR models to a real data set D, referring to the number of cases of dengue fever recorded in the city of Campo Grande, Mato Grosso do Sul state, Brazil, in the period from January 2008 to December 2019. The dataset D is an excel sheet composed of 144 lines and 6 columns. The first column contains the recorded number of dengue-fever cases in each of the 144 months considered in the study. Columns 2 to 6 contain the recorded values for the following explanatory variables: month, the average temperature in the month, the average humidity in the month, the number of rainy days in the month, and rainfall in the month.

To estimate the model parameters, we adopt the maximum-likelihood method. Since the maximum-likelihood estimators do not have explicit mathematical solutions, we obtain the estimates numerically by using the R software [13] and the command gamlss of the Generalized Additive Model for Location, Scale and Shape (GAMLSS) package [14]. According to Stasinopoulos et al. [15], “the GAMLSS were introduced by Rigby and Stasinopoulos (2001, 2005) [14,16] and Akantziliotou et al. (2002) [17] as a way of overcoming some of the limitations associated with Generalized Linear Models (GLM) and Generalized Additive Models (GAM)”. The two main advantages of a GAMLSS model are: (i) it assumes that the response (dependent) variable may follow any parametric distribution and not just distributions belonging to an exponential family, and (ii) all the parameters of the probability distribution of the response variable can be modelled as functions of the available explanatory variables. More details on GAMLSS package may be found in its manual available on the website http://www.gamlss.com/wp-content/uploads/2013/01/gamlss-manual.pdf (accessed on 15 March 2022).

We also compare the performance of the NBR and PIGR models by using the Akaike Information criterion [18,19], denoted by AIC, and the Bayesian Information criterion [20], denoted by BIC, and the Root Mean Square Error (RMSE). We also fit both models by considering a P-spline term for the month variable since it has cyclical values, and smooth terms for continuous variables. For this, we use the pbc() and pb() functions inside the gamlss function. Based on the AIC, BIC and RMSE values, the PIGR model was considered the best model. We also present the quantile-quantile normal plot and worm plot for the randomized quantile residuals [21] generated from the NBR and PIGR fitted models. Both graphs also show PIGR performing better than NBR.

The three main advantages of the proposed modeling are: (i) present better performance in relation to the usual approaches, which are based on the fitting of PR and NBR models; (ii) the fitted model shows that every year a peak will occur, and that the only way to avoid this peak is by the implementation of actions to combat the proliferation of the transmitting mosquito; and (iii) the fitted model shows in which the months of the year combat actions must be implemented.

The remainder of the paper is organized as follows. In Section 2, we describe the PR, NBR and PIGR models and present the estimation procedure. Section 3 presents the main results, including the comparison of the NBR and PIGR models and the residual analysis. Section 4 presents the final remarks.

## 2. Statistical Modeling

Let $y=(y_{1},\ldots ,y_{n})$ be a vector of data composed of the number of dengue-fever cases recorded in a period of *n* months in a country, state, or city. Assume that recorded value $y_{t}$ is a realization of the random variable $Y_{t}$, for $Y_{t}\in Y=\left\{0,1,2,3,\ldots \right\}$.

In addition, assume that measurements of *p* explanatory variables are available, denoted by $X_{1},\ldots ,X_{p}$, that can be associated with mosquito reproduction and dengue transmission, and consequently also associated with the number of recorded cases of dengue. Consider x to be an n×(p+1) matrix in which the first column contains only values 1 and columns 2 to p+1 are composed of the recorded measurements for variables $X_{1}$ to $X_{p}$, respectively. Denote the $t^{\mathrm{th}}$ line of x by $x_{t}=\left(1,x_{t1},\ldots ,x_{tp}\right)$, for t=1,…,n.

### 2.1. Poisson Regression Model

Since random variable $Y_{t}$ is a discrete variable that counts the number of cases in a time period of one month *t*, it is usual to assume that $Y_{t}$ follows a Poisson distribution with parameter $\mu_{t}$, i.e.,
$$Y_{t}∼\mathrm{Poisson}\left(\mu_{t}\right),$$
where, $\mu_{t}=E\left(Y_{t}\right)$ is the expected value of $Y_{t}$, with $\mu_{t}>0$, t=1,…,n. Its probability mass function is given by
$$P\left(Y_{t}=y_{t}|\mu_{t}\right)=\frac{\mu_{t}^{y_{t}}e^{-\mu_{t}}}{y_{t}!},$$
for $y_{t}\in Y$ and t=1,…,n.

Using the theory of generalized linear models [5], we can link the expected value of $Y_{t}$ to explanatory variables x through the following log-linear relationship:(1)$$\eta \left(\mu_{t}\right)=\log\left(\mu_{t}\right)=\beta x_{t}=\beta_{0}+\sum_{j=1}^{p}\beta_{j}x_{tj},$$
where $\eta (\mu_{t})$ is the linear predictor, $\beta ={\left(\beta_{0},\beta_{1},\ldots ,\beta_{p}\right)}^{\prime }$ is the vector of parameters of the model and $x_{t}$ is the *t*-th line of the matrix x, for t=1,…,n.

Given (y,x) the log-likelihood function for parameters β is given by
$$l\left(\beta |y,x\right)\propto \sum_{t=1}^{n}y_{t}\beta x_{t}-\exp\left\{\beta x_{t}\right\}.$$

In order to get the maximum-likelihood estimates for the parameters β, we first need to determine the first-order partial derivatives of the log-likelihood function, which are given by
(2)$$\frac{\partial l(\beta )}{\partial \beta_{j}}=\sum_{t=1}^{n}x_{tj}\left(y_{t}-\exp\left\{\beta_{0}+\sum_{j=1}^{p}\beta_{j}x_{tj}\right\}\right),$$
for j=0,…,p.

The maximum-likelihood estimates are the solutions of equations in (Equation 2) when they are set to 0, $\frac{\partial l(\beta )}{\partial \beta_{j}}=0$, for j=0,…,p. However, these equations do not have explicit analytic solutions. Therefore, we apply numerical methods to solve these equations. We can obtain the maximum-likelihood estimates $\hat{\beta }$ of the parameters β using the R software [13] and the function glm() [22].

Although the Poisson distribution is a natural choice for modeling the number of dengue-fever cases recorded in a month, this distribution has the restriction that the expected value is equal to the variance, $E\left(Y_{t}\right)=Var\left(Y_{t}\right)$, for t=1,…,n. Thus, before considering a Poisson regression model it is essential to check if recorded data present some evidence for overdispersion or underdispersion.

Hinde and Demétrio [23] propose to check the evidence for overdispersion or underdispersion by using the index
(3)$$\mathrm{IS}=\frac{S_{y}^{2}-\bar{y}}{\bar{y}},$$
where $S_{y}^{2}$ and $\bar{y}$ are the sampled variance and mean of the recorded values for *Y*, respectively. The decision is based on the following interpretation: If IS=0, the recorded data indicate equidispersion, and the Poisson regression model can be used. On the other hand, if IS<0, the recorded data indicate underdispersion, and if IS>0, the recorded data indicate overdispersion. The Poisson regression model is not appropriate for nonzero IS.

Cameron and Trivedi [24] propose to check for evidence of overdispersion using a hypothesis test. To do this, the authors assume that $\mathrm{Var}\left(Y\right)=\mu +\lambda \mu^{2}$, and specify the following statistical hypotheses $H_{0}:\lambda =0$
*versus*
$H_{1}:\lambda >0$. The test statistic is calculated according to the following four steps:(i)Fitting a Poisson regression model;(ii)Calculating the fitted values ${\hat{\mu }}_{t}$, for t=1,…,n;(iii)Calculating the auxiliary values
$$Y_{t}^{*}=\frac{{\left(y_{t}-{\hat{\mu }}_{t}\right)}^{2}-y_{t}}{{\hat{\mu }}_{t}},\;\;\mathrm{for}\;\;t=1,\ldots ,n;$$(iv)Fitting of an auxiliary linear model $Y_{t}^{*}=\lambda {\hat{\mu }}_{t}+\varepsilon_{t}$, where $\varepsilon_{t}$ is a random error, for t=1,…,n.

According to Cameron and Trivedi [24], the t-statistic for λ is asymptotically normal under the null hypothesis of no overdispersion. The null hypothesis is rejected whenever the *p*-value associated with the calculated statistic is smaller than a significance level α, with 0<α<1. This overdispersion test may be performed in the R software using the overdisp() function of the overdisp package [25]. For overdispersed data, an alternative is to consider the negative binomial regression model.

### 2.2. Negative Binomial Regression Model

Assume $Y_{t}$ follows the negative binomial distribution with parameters $\mu_{t}$ and ν,
$$Y_{t}∼\mathrm{NB}\left(\mu_{t},\nu \right),$$
for $\mu_{t}>0$, ν>0, $Y_{t}\in Y$ and t=1,…,n.

According to [24], the negative binomial distribution that accomodates overdispersion in the data has the following probability mass function:$$P\left(Y_{t}=y_{t}|\mu_{t},\nu \right)=\frac{\Gamma \left(y_{t}+\nu^{-1}\right)}{\Gamma \left(\nu^{-1}\right)\Gamma \left(y_{t}+1\right)}{\left(\frac{\nu^{-1}}{\nu^{-1}+\mu_{t}}\right)}^{\nu^{-1}}{\left(\frac{\mu_{t}}{\nu^{-1}+\mu_{t}}\right)}^{y_{t}}$$
where Γ(·) is the gamma function. The expected value and variance of $Y_{t}$ are given by $E\left(Y_{t}\right)=\mu_{t}$ and $\mathrm{Var}\left(Y_{t}\right)=\mu_{t}+\nu \mu_{t}^{2}$, respectively, for t=1,…,n.

As it is in the PR model, the expected value for $Y_{t}$ in the NBR model is linked to the explanatory variables X via a function of the form given in expression  (Equation 1). The log-likelihood function for the parameters (β,ν) is
$$l\left(\beta ,\nu |y,x\right)\propto \sum_{t=1}^{n}A_{t}\left(\nu \right)-\frac{1}{\nu }\log\left(1+\nu \exp\left\{\beta x_{t}\right\}\right)+y_{t}\beta x_{t}-y_{t}\log\left(1+\nu \exp\left(\beta x_{t}\right)\right)+y_{t}\log\left(\nu \right),$$
where $A_{t}\left(\nu \right)=\log\left(\Gamma \left(y_{t}+\frac{1}{\nu }\right)\right)-\log\left(\Gamma \left(\frac{1}{\nu }\right)\right)$, for t=1,…,n.

The maximum-likelihood estimates are obtained by determining the first-order partial derivatives of the log-likelihood function, then equating them to zero:$$\begin{array}{ccc} \frac{\partial l(\beta ,\nu )}{\partial \beta_{j}} & = & \sum_{t=1}^{n}x_{t(j+1)}\left(\frac{y_{t}-\exp\left\{\beta x_{t}\right\}}{1+\nu \exp\left\{\beta x_{t}\right\}}\right)=0;\\ \frac{\partial l(\beta ,\nu )}{\partial \nu } & = & \sum_{t=1}^{n}\left[\frac{\partial A_{t}\left(\nu \right)}{\partial \nu }+\frac{y_{t}}{\nu }-\left(y_{t}-\frac{1}{\nu }\right)\frac{\exp\left\{\beta x_{t}\right\}}{1+\nu \exp\left\{\beta x_{t}\right\}}-\frac{1}{\nu^{2}}\log\left(1+\nu \exp\left\{\beta x_{t}\right\}\right)\right]=0. \end{array}$$
for j=0,…,p.

These equations also do not have explicit solutions. Analogously to the case of the Poisson regression model, we obtain the maximum-likelihood estimates $\left(\hat{\beta },\hat{\nu }\right)$ of the parameters $\left(\beta ,\nu \right)$ using the R software, but for this case we use the gamlss() function from the gamlss package [26] with the option family=NBI.

### 2.3. Poisson-Inverse-Gaussian Regression Model

As an alternative to the negative binomial model, consider that $Y_{t}$ follows the Poisson-inverse Gaussian distribution with parameters $\mu_{t}$ and τ, i.e., 
$$Y_{t}∼\mathrm{PIG}\left(\mu_{t},\tau \right),$$
for t=1,…,n. This distribution is a mixture of a Poisson distribution and an inverse Gaussian distribution. Let $Y_{t}|V$ follow a Poisson distribution with mean $\mu_{t}V$, where *V* follows an Inverse Gaussian distribution with mean equal to 1 and dispersion parameter 1/τ [8]. The marginal probability mass function for $Y_{t}$ is
$$P\left(Y_{t}=y_{t}|\mu_{t},\tau \right)=\frac{\mu_{t}^{y_{t}}}{y_{t}!}{\left(\frac{2}{\pi \tau }\right)}^{0.5}\exp\left\{1/\tau \right\}{\left(1+2\tau \mu_{t}\right)}^{-S_{t}/2}K_{S_{t}}\left(\Psi_{t}\right),$$
where $S_{t}=y_{t}-\frac{1}{2}$, $\Psi_{t}=\frac{\sqrt{1+2\tau \mu_{t}}}{\tau }$ and $K_{S_{t}}\left(\Psi_{t}\right)$ is the modified Bessel function of second kind [11], for t=1,…,n.

Considering the link function given in Equation (Equation 1), the log-likelihood function for parameters (β,τ) is given by
$$l\left(\beta ,\tau |y,x\right)\propto \sum_{t=1}^{n}y_{t}\beta x_{t}-\frac{\log(\tau )}{2}+\frac{1}{\tau }-\frac{S}{2}\log\left(1+2\tau \exp\left\{\beta x_{t}\right\}\right)+\log\left(K_{S_{t}}\left(\Psi_{t}\right)\right).$$

Settting the first-order partial derivatives of the log-likelihood function equal to zero, we obtain
$$\begin{array}{ccc} \frac{\partial l(\beta ,\tau |y,x)}{\partial \beta_{j}} & = & \sum_{t=1}^{n}\left[x_{tj}\left(y_{t}-\frac{\tau S_{t}\exp\left\{\beta x_{t}\right\}}{1+2\tau \exp\left\{\beta x_{t}\right\}}\right)+\frac{\partial \log\left(K(\Psi_{t})\right)}{\partial \beta_{j}}\right]=0;\\ \frac{\partial l(\beta ,\tau |y,x)}{\partial \tau } & = & -\frac{n}{\tau^{2}}-\frac{n}{2\tau }-\sum_{t=1}^{n}\left(\frac{S_{t}\exp\left\{\beta x_{t}\right\}}{1+2\tau \exp\left\{\beta x_{t}\right\}}+\frac{\partial \log\left(K(\Psi_{t})\right)}{\partial \tau }\right)=0, \end{array}$$
for j=0,…,p.

Since the maximum-log-likelihood equations are nonlinear, they cannot be solved analytically. Therefore, we obtain the maximum-likelihood estimates $\left(\hat{\beta },\hat{\tau }\right)$ of the parameters $\left(\beta ,\tau \right)$ using the R software and the gamlss package’s gamlss() function, with the option family = PIG.

#### Simulation Study for PIGR Model

Since the main focus of this article is to describe the performance of the PIGR model, in this section we present a simulation study that illustrates the performance of this model. For this purpose, we generated values $Y_{t}$ from a PIG distribution with parameters $\mu_{t}=\exp\left(\beta_{0}+\beta_{1}x_{1t}+\beta_{2}x_{1t}\right)$ and τ=1, for t=1,…,n. The sample sizes considered were n={50,100,150,200}. We set $\beta_{0}=1.5$, $\beta_{1}=1.5$ and $\beta_{2}=-1$ and generate values for covariates $X_{1}$ and $X_{2}$ from the following normal distributions, $X_{1t}∼N\left(0,1\right)$ and $X_{2t}∼N\left(4,1\right)$, for t=1,…,n.

In order to verify the frequentist properties of the maximum-likelihood estimator (MLE) $\hat{\theta }=\left({\hat{\beta }}_{0},{\hat{\beta }}_{1},{\hat{\beta }}_{2},\hat{\tau }\right)$ for the parameters of the PIGR model, we generate B=1000 different artificial datasets for each sample size *n* and summarize the results in terms of the average of estimates, bias, and mean square error (MSE). Table 1 shows these values for each of the parameters. As one can note, as sample size increases, there is a reduction in the bias and MSE values. These results show us empirically that there is no reason for doubting that the ML estimator $\hat{\theta }$ satisfies the asymptotic properties of MLEs [27], i.e., $\hat{\theta }$ is asymptotically consistent, unbiased, and is approximately a normal random variable.

## 3. Application

In this section, we apply the PR, NBR and PIGR models to a real data set containing the number of dengue-fever cases recorded in the city of Campo Grande, state of Mato Grosso do Sul, Brazil, in the period from January 2008 (t=1) to December 2019 (t=144).

The city of Campo Grande is located in the transition zone between a humid mesothermal climate without drought and a humid tropical climate, with a rainy season in the summer and a dry season in the winter. The city has its climate controlled by three characteristic air masses: the Atlantic Polar Mass, coming from the south, the Continental Equatorial Mass, coming from the north and the Continental Tropical Mass, which forms in the lower Chaco region. The rainy season runs from October to March. The average total annual precipitation is 1225 mm. The relative humidity of the air presents values close to 80% from December to February. From March onwards, relative humidity shows a gradual decline, reaching its minimum value of approximately 60% in August. From August onwards, the relative humidity of the air rises again. The average maximum temperature is around $25{\;}^{{}^{\circ}}$C in the period from October to March [28].

Due to the favorable climate for the proliferation of the dengue-transmitting mosquito, especially, between October and March, the city has a large number of dengue cases recorded every year. A dengue-control strategy implemented by the city government is based on the availability of health agents in city neighborhoods to provide information on dengue and how to eliminate the transmitter mosquito. Additionally, the city government has a program for cleaning neighborhoods to eliminate possible breeding sites of the dengue-transmitting mosquito.

Thus, in order to contribute to the dengue surveillance system in the city of Campo Grande—MS, this article proposes the fitting of a statistical model to identify the climatic variables that can influence the number of dengue cases. Once the variables are identified, the fitted model allows projections to and simulation of different scenarios of evolution of the number of cases of the disease. Therefore, it can help in decision-making regarding the implementation of measures to combat and/or control the vector that transmits the disease.

### Results

Consider $y=(y_{1},\ldots ,y_{n})$ to be the number of dengue-fever cases recorded in the city of Campo Grande, MS state, Brazil, in the period from January 2008 (t=1) to December 2019 (t=164). These measures are freely available on the website http://tabnet.datasus.gov.br/cgi/tabcgi.exe?sinannet/cnv/denguebbr.def (accessed on 10 November 2020) and also can be obtained by emailing the authors of the present article.

Let x be a matrix of dimension n×5 composed of the recorded measures of the variables
(4)$$\begin{array}{ccc} X_{1} & : & \;\mathrm{Month}\;\mathrm{of}\;\mathrm{the}\;\mathrm{year},\;\mathrm{coded}\;\mathrm{from}\;1\;\mathrm{to}\;12;\\ X_{2} & : & \;\mathrm{average}\;\mathrm{temperature}\;\mathrm{in}\;\mathrm{the}\;\mathrm{month};\\ X_{3} & : & \;\mathrm{average}\;\mathrm{humidity}\;\mathrm{in}\;\mathrm{the}\;\mathrm{month};\\ X_{4} & : & \;\mathrm{number}\;\mathrm{of}\;\mathrm{rainy}\;\mathrm{days}\;\mathrm{in}\;\mathrm{the}\;\mathrm{month};\\ X_{5} & : & \;\mathrm{rainfall}\;\mathrm{in}\;\mathrm{the}\;\mathrm{month}. \end{array}$$

The recorded measures for variables $X_{2}$ to $X_{5}$ are freely available at https://www.cemtec.ms.gov.br (accessed on 8 December 2020). Denote this dataset by D=(y,x), which is a matrix of dimension n×6. The first column contains the recorded number of dengue-fever cases in each of the 144 months considered in the study. Columns 2 to 6 contain the recorded values of the explanatory variables $X_{1}$ to $X_{5}$.

Figure 1 shows the number of recorded dengue-fever cases from 2007 to 2019. The figure includes the number of cases recorded in 2007 just to show that every three years the city of Campo Grande presents a larger outbreak of dengue-fever cases. However, the recorded number of dengue-fever cases in 2007 was not considered to fit the models because the website https://www.cemtec.ms.gov.br (accessed on 8 December 2020) does not include values for the explanatory variables $X_{2}$ to $X_{5}$ in 2007.

Figure 2 shows the evolution of the number of dengue-fever cases by month. Due mainly to the climate of the city, characterized by high heat and heavy rains from October to March, this period contains most of the recorded dengue-fever cases in the city. This fact shows the importance of having a model for projection for the number of dengue cases from environmental variables, to support actions to combat the proliferation of the mosquito and consequently the reduction of the number of cases.

Table 2 shows the descriptive statistics of the recorded *y* values in the period from January of 2008 to December of 2019. The smallest recorded value was 2 cases in August of 2008. The highest recorded value was 18,530 cases in January of 2013. On average, 1057 cases were recorded per month in the period considered.

Table 3 shows the correlations for each pair of variables. As one can note, the highest correlation is between variables $X_{3}$ and $X_{5}$. However, since it is not a strong correlation (>0.75), we opt to maintain both variables for the fitting of the models.

In addition, we also verify if there is multicollinearity among explanatory variables by means of variance inflation factor (VIF) values for the PR and NBR models [29]. At this point, we remind the reader that multicollinearity occurs when two or more explanatory variables are highly correlated with one another in a regression model. That is, one explanatory variable can be predicted from another expanatory variable. A VIF value equal to 1 means that the predictor is not correlated with other variables. The higher the value, the greater the correlation of the variable with other variables. In general, values smaller than 5 indicate weak correlation, values between 5 and 10 indicate moderate correlation, and values equal to or greater than 10 indicate high correlation.

In order to calculate the VIF values, we first fit the PR and NBR models using the R software and the glm function. We then obtain the VIF values by applying the vif function of the car package. Listing 1 shows the R code used. The VIF values are presented in Table 4. As one can see, all values are less than five, which indicates weak multicollinearity. Therefore, all five explanatory variables are used to fit the models.    

**Listing 1.** R code.

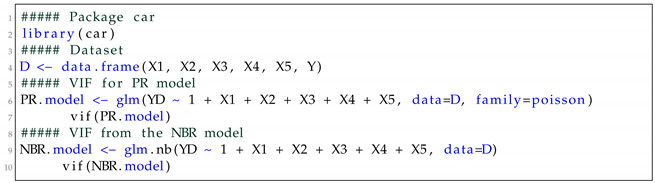



Using the sample average and sample variance presented in Table 2, the overdispersion index given in Expression (Equation 3) is $\mathrm{IS}=\frac{7269590-1057}{1057}=6876.614$. That is, the recorded values are overdispersed. Additionally, we also apply the overdisperion test of Cameron and
Trivedi [24] (CT test), using the overdisp() function of the R software. Figure 3 shows the output of the test in the R software. As one can note, the null hypothesis is rejected for the usual significance levels α={0.10,0.05,0.01}, meaning that there is evidence for overdispersion.

Both results described above indicate that the PR model is not appropriate for this dataset. Due to this, hereafter we fit the NBR and PIGR models to the dataset and compare these two models according to the AIC and BIC model-selection criteria. The best model is the one that has the smallest AIC and BIC values.

We fit NBR and PIGR models using the gamlss() function of the gamlss package of the R software. Since the month variable has cyclical values, we fit both models by considering a cyclical P-spline term for this variable. For this, we use the pbc() function inside the gamlss function. In addition, we fit both models by considering smooth terms for continuous variables $X_{2}$, $X_{3}$ and $X_{5}$. For this case, we use the pb() function. We call the models fitted with pb() function of NBR-S and PIGR-S, respectively. Listing 2 shows the R code used for fitting the models.   

**Listing 2.** R code.

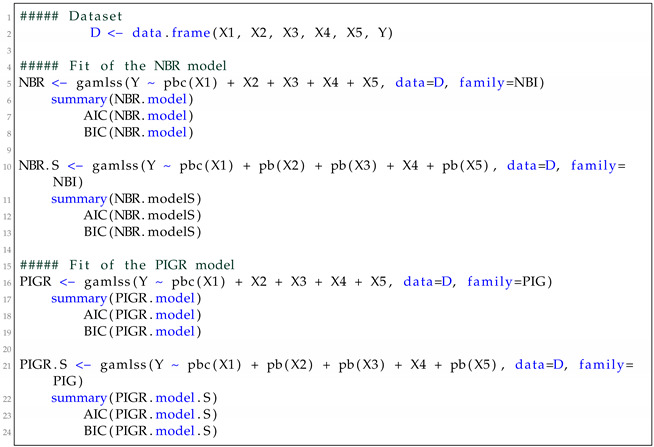



To significance level α=0.10, none of the variables was significant for the NBR model (*p*-value >α). For NBR-S and PIGR models, variables $X_{4}$ and $X_{5}$ were not significant (*p*-values >α). For the PIGR-S model, $\beta_{0}$ and the variables $X_{4}$ and $X_{5}$ were not significant (*p*-values >α). Due to this, we discard the NBR model and refit NBR-S, PIGR, and PIGR-S models without the non-significant variables.

Table 5 shows model-comparison criteria for the three fitted models. The smallest values are highlighted in boldface. Since the AIC and BIC values for the PIGR and PIGR-S models are very similar and the RMSE values are equal, we opt to maintain the PIGR as the best model because the smooth terms have not led to a significant improvement in the model.

With the models fitted, it is important to perform a residuals analysis in order to identify the discrepancies between the models and the data, and to assess the overall model goodness-of-fit. In a normal linear regression scenario, the Pearson and deviance residuals are usually considered. However, these residuals are not suitable for problems in which the response variable is discrete because they are not normally distributed, and according to Feng et al. [30], “have nearly parallel curves according to the distinct discrete response values, imposing great challenges for visual inspection”. To circumvent this issue, Dunn
and Smyth [21] propose the use of randomized quantile residuals (RQR). According to the authors, this kind of residuals is particularly ideal for visualizing the goodness-of-fit of count regression models.

In order to calculate the RQR, we first need to obtain the cumulative distribution function, $F(y_{t}|{\hat{\mu }}_{t},\hat{\tau })$ of the model considered, for t=1,…,n. For the continuous case, F(·) values are uniformily distributed on interval (0,1), and the RQR is defined as $r_{t}=\Phi^{-1}\left(F\left(y_{t}|{\hat{\mu }}_{t},\hat{\tau }\right)\right),$ where Φ(·) is the cumulative distribution function of the standard normal distribution. However, since the cumulative distribution function F(·) for the models considered (NBR and PIGR) is not strictaly continuous, but a step function, a randomization is introduced to produce continuous normal residuals. Thus, in order to get the RQR, Dunn and Smyth [21] propose the following strategy. For t=1,…,n:Determine a point $a_{t}=\lim_{y↑y_{t}}F\left(y_{t}|{\hat{\mu }}_{t},\hat{\tau }\right)$, i.e., $a_{t}$ is the value of F(·) when approaching $y_{t}$ from the left;Determine $b_{t}=F\left(y_{t}|{\hat{\mu }}_{t},\hat{\tau }\right)$, i.e., the value of F(·) at the point $y_{t}$;Generate a value $u_{t}$ from a uniform distribution on interval $\left(a_{t},b_{t}\right]$;Calculate the RQR ${\hat{r}}_{t}=\Phi^{-1}\left(u_{t}\right)$.

We obtained the RQR values for NBR and PIGR models using the residuals function of the R software.

Figure 4 shows the normal quantile-quantile plot (q-q plot) for the randomized quantile residuals of the NBR-S and PIGR fitted models. The q-q plot is a scatterplot created by plotting the empirical quantiles of the residuals against the theoretical quantiles of the normal distribution. If residuals are normally distributed then they should form an approximately straight line. Figure 5 shows the worm plot. This graph was proposed by van Buuren and Fredriks [31] to identify regions (intervals) of the explanatory variable within which the model does not fit the data adequately [15]. In this graph, the upwardsline of the q-q plot is rotated to the horizontal in order to remove the trend and the *Y* axis contains the difference between its location in the theoretical and empirical distributions. If the residuals follow a normal distribution then the *Y* values are near the horizontal line and consequently inside the confidence band. The R function wp() provides the worm plot for a gamlss fitted model. As one can note, both figures indicate the PIGR model performs better than the NBR model. In addition, the graphs of the residuals from the PIGR model indicate that there is no reason to worry about the inadequacy of the fit. Table 6 shows the estimates for the parameters of the PIGR model.

Figure 6 shows estimated relationships between the response variable and explanatory variables. As expected, the relationship with $X_{1}$ (month) presents a cyclical behavior, and the relationship with $X_{2}$ and $X_{3}$ is linear. These graphs were constructed using the term.plot function of the R software.

Figure 7 shows the number of registered dengue cases (symbol •) and a confidence band of 95% generated from the fitted PIGR model. In order to construct the confidence band we use a parametric bootstrap. That is, from estimated value ${\hat{\mu }}_{t}$ and $\hat{\tau }$, we generate L=1000 values from a PIG distribution using the rPIG function of the R software. Then we set the lower and upper limits as being the percentiles 2.5% and 97.5% of the generated values. As one can note, the fitted model indicates that every year a peak will occur. How high or low the recorded number of dengue cases will be in relation to the expected peak (given by the fitted model) is controlled by action taken to combat the proliferation of the mosquito. If such action is effective, there is no occurence of a peak, as in years 2008, 2009, 2011, 2012, 2014 and 2017. Otherwise, the peak may be higher than expected, i.e., there may be a larger outbreak, as in the years 2010, 2013, 2016 and 2019. That is, human behavior has a great influence on the number of cases that will be recorded. However, since this behavior is very difficult to quantify and is not present in the proposed model, this also has an influence on the predictive performance of the fitted model.

For example, in the next year after the years with peaks of cases (2007, 2010, 2013 and 2016), there was a significant reduction of recorded cases due to the implementation of actions to combat the proliferation of the disease vector and awareness campaigns reminding the population what happened the previous year. However, with the expected reduction in the number of recorded dengue cases obtained, the combat actions and awareness campaigns were not maintained, leading to an increase in the number of cases in the following two years. This has been occurring cyclically over the last 13 years.

Thus, although the proposed model does not present a satisfactory predictive performance, especially due to our inability to quantify and insert into the model the actions taken to combat the transmitting mosquito, it has at least three advantages: (i) better performance in comparison to the usual approaches, which are based on the fitting of PR and NBR models; (ii) the fitted model shows that a peak will occur every year and that the only way to avoid this peak is via the implementation of actions to combat the proliferation of the transmitting mosquito; and (iii) the fitted model shows which are the months of the year in which combat actions must be implemented.

## 4. Final Remarks

Dengue is a disease that affects millions of people every year, especially in tropical nations, causing a great impact on public health systems. Due to this, there is an interest in the development of statistical models that can forecast the number of dengue-fever cases and also identify which environmental variables may be related to the number of recorded cases.

In this article, we present statistical modeling for an overdispersed, long-tailed dengue-fever dataset. The proposed modeling is based on the assumption that the recorded number of dengue-fever cases in a month is generated according to a Poisson-inverse-Gaussian distribution. According to Zhu and Joe [7], this distribution may be used for modeling overdispersed, long-tailed datasets and presents a larger range of skewness than negative binomial distribution.

We model the expected number of dengue-fever cases as being linked to a set of explanatory variables through a log-linear function. This approach is called a Poisson-inverse-Gaussian regression (PIGR) model. In order to estimate the parameters of interest, we adopt the maximum-likelihood method. Since the estimators do not have known analytic solutions, we obtain estimates numerically by using the gamlss() function of the gamlss package of the R software.

We compare the proposed modeling to the usual approach based on the use of a negative-binomial regression (NBR) model. The two models were compared by using the AIC, BIC and RMSE criteria. Additionally, we compare the two models in terms of randomized quantile residuals. Three model-selection criteria indicate the PIGR model as better than the NBR model for this application. The randomized quantile residuals also indicate that PIGR performs better than NBR. That is, it has a quantile-quantile normal plot with residuals near the line y=x, and a worm plot with residuals near the horizontal line.

The fitted PIGR model indicates that variables $X_{1}$ (month), $X_{2}$ (temperature) and $X_{3}$ (humidity) are related to the recorded number of dengue-fever cases. Variables $X_{2}$ and $X_{3}$ are positively related to the number of dengue cases. That is, an increase of the temperature and/or the humidity in the air is expected to lead to an increase in the recorded number of dengue cases. This makes intuitive sense because these two variables are directly related to favorable conditions for the development of the mosquito that transmits dengue fever. According to Silva et al. [32] “the female mosquito, infected and subjected to temperatures of approximately $32{\;}^{{}^{\circ}}$C, has 2.64 times more chance of completing the incubation period than those subjected to mild temperatures”.

As a final result, the fitted model expects a peak in cases every year (see Figure 7). Based on this result, we conjecture the only way to avoid the peak is through an intervention by humans which avoids the proliferation of the transmiting mosquito. All analyses were performed using the R software and source code can be obtained by emailing the authors.

To end the paper we highlight the following three points: (i) Although the proposed modeling has presented better performance than usual approaches, it shares the basic assumptions of the Poisson and negative binomial regression models, which are: log-linearity in model parameters and independence of individual observations; (ii) as pointed out before, the main advantage is to be able to model overdispersed long-tail datasets; (iii) However, since dengue fever data are recorded longitudinally, they may present some kind of temporal correlation. Thus, the development of a modeling approach that incorporates correlation among recorded values for the answer variable can be viewed as future work. An approach we are currently studying is the development of a PIGR mixed-effect model.

## Figures and Tables

**Figure 1 entropy-24-01256-f001:**
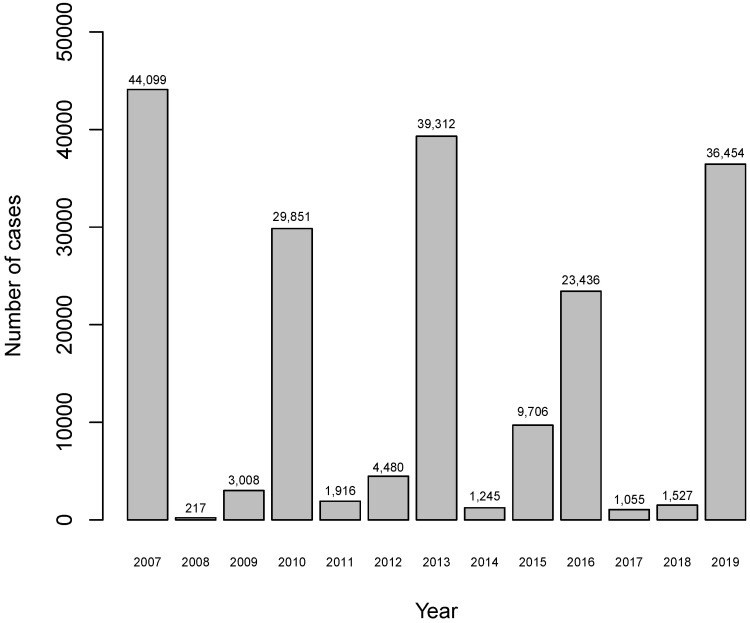
Number of recorded dengue-fever cases by year from 2007 to 2019.

**Figure 2 entropy-24-01256-f002:**
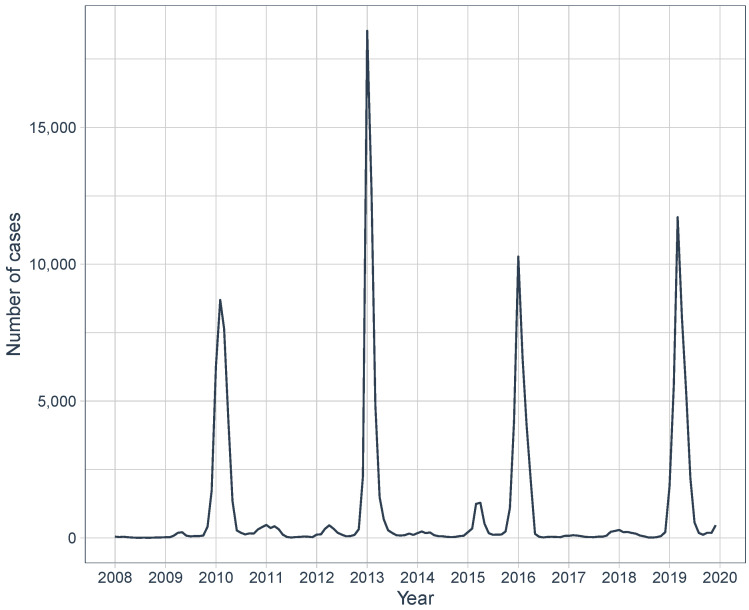
Evolution of the number of dengue-fever cases by month in the period considered (January 2008 to December 2019).

**Figure 3 entropy-24-01256-f003:**
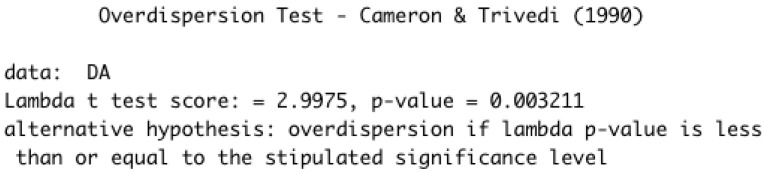
Outputs of the CT test for overdispersion using the overdisp() function.

**Figure 4 entropy-24-01256-f004:**
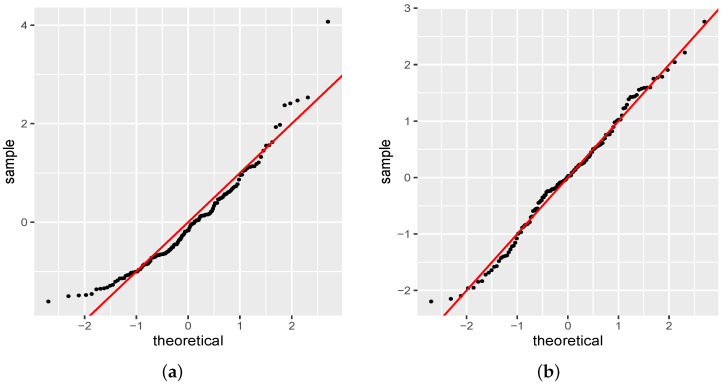
Normal quantile-quantile plot for the residuals. (**a**) NBR model. (**b**) PIGR model.

**Figure 5 entropy-24-01256-f005:**
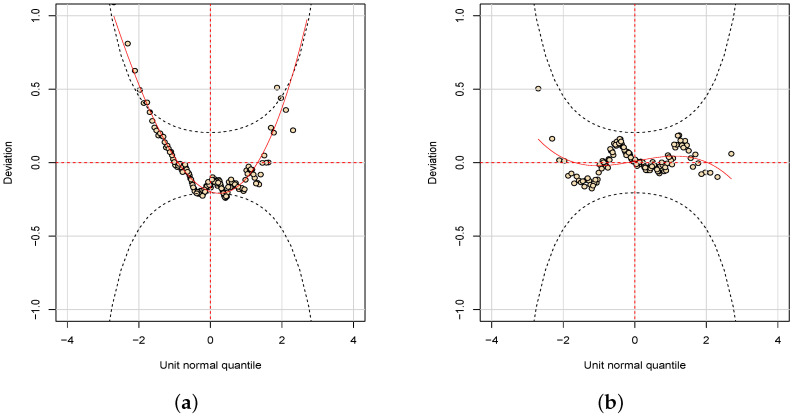
Worm plot. (**a**) NBR model. (**b**) PIGR model.

**Figure 6 entropy-24-01256-f006:**
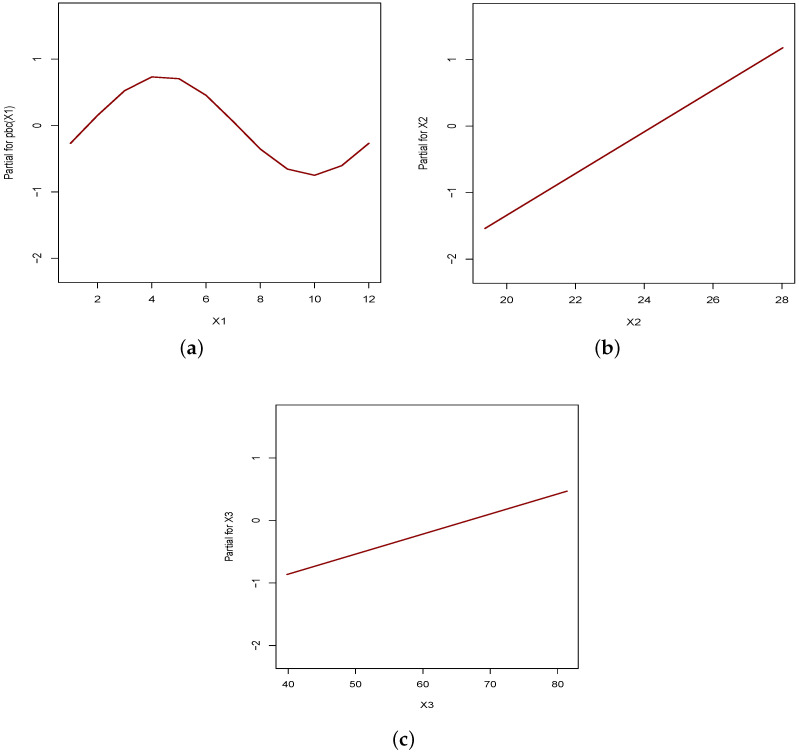
Estimated relationship between response variables and explanatory variables. (**a**) Relationship with $X_{1}$. (**b**) Relationship with $X_{2}$. (**c**) Relationship with $X_{3}$.

**Figure 7 entropy-24-01256-f007:**
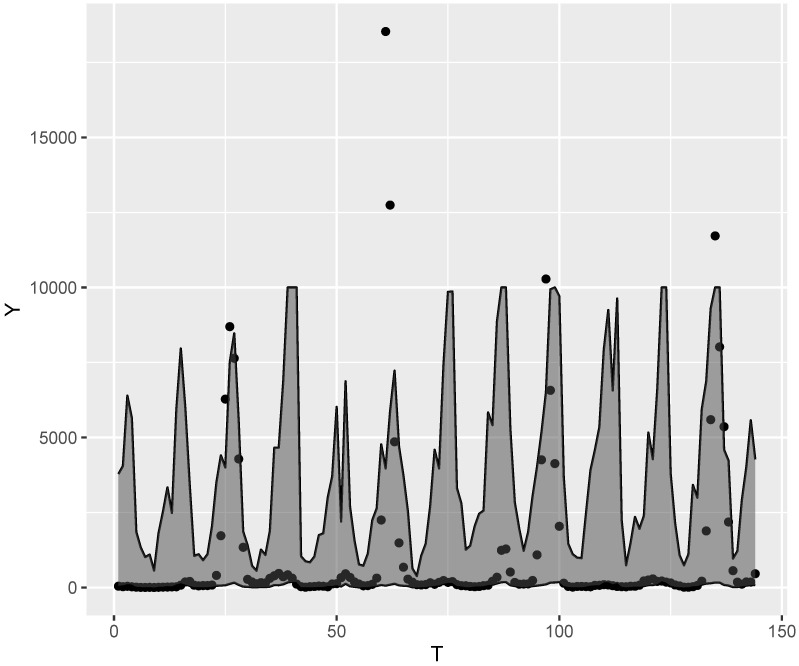
Recorded values and confidence band (95%) generated from fitted model.

**Table 1 entropy-24-01256-t001:** Average of estimates, bias and mean square error (MSE) values.

	n=50				n=100		
Parameter	Average	BIAS	MSE	Parameter	Average	BIAS	MSE
$\beta_{0}$	1.8198	0.3198	14.1991	$\beta_{0}$	1.4642	−0.0357	0.2001
$\beta_{1}$	−1.9400	−0.4400	7.9899	$\beta_{1}$	−1.5418	−0.0418	0.2140
$\beta_{2}$	−0.6580	−0.1580	1.3534	$\beta_{2}$	−0.5136	−0.0136	0.0192
τ	1.2681	0.2681	0.4752	τ	1.1292	0.1292	0.2048
	n=150				n=200		
**Parameter**	**Average**	**BIAS**	**MSE**	**Parameter**	**Average**	**BIAS**	**MSE**
$\beta_{0}$	1.4623	−0.0376	0.0896	$\beta_{0}$	1.4541	−0.0458	0.0579
$\beta_{1}$	−1.5777	−0.0177	0.0978	$\beta_{1}$	−1.4972	0.0027	0.0559
$\beta_{2}$	−0.5036	−0.0036	0.0082	$\beta_{2}$	−0.4982	0.0017	0.0055
τ	1.0774	0.0774	0.0843	τ	1.641	0.0641	0.0594

**Table 2 entropy-24-01256-t002:** Descriptive statistics of the recorded numbers of dengue-fever cases.

Minimum	1st Quartile	Median	Average	3rd Quartile	Maximum	Variance
2	45.5	124.5	1057	340.2	18,530	7,269,590

**Table 3 entropy-24-01256-t003:** Correlations.

Variables	$X_{1}$	$X_{2}$	$X_{3}$	$X_{4}$	$X_{5}$
$X_{1}$	1.00	0.09	−0.11	0.17	−0.39
$X_{2}$	0.09	1.00	0.33	0.09	0.20
$X_{3}$	−0.11	0.33	1.00	−0.14	0.59
$X_{4}$	0.17	0.09	−0.14	1.00	−0.19
$X_{5}$	−0.39	0.20	0.59	−0.19	1.00

**Table 4 entropy-24-01256-t004:** VIF values.

Model	$X_{1}$	$X_{2}$	$X_{3}$	$X_{4}$	$X_{5}$
PR	1.4549	1.1254	2.2633	1.5898	2.5500
NBR	1.1998	1.5677	3.2224	2.5796	4.4537

**Table 5 entropy-24-01256-t005:** Model-comparison criteria.

Model	AIC	BIC	RMSE
NBR-S	2044	2059	2885
PIGR	2001	2016	**2560**
PIGR-S	**1999**	**2011**	**2560**

**Table 6 entropy-24-01256-t006:** Estimates for parameters of PIGR model.

Parameter	Estimate	Str. Dev.	*p*-Value
$\beta_{0}$	−3.3276	1.5920	0.0384
$\beta_{2}$	0.313 8	0.0596	<0.0001
$\beta_{3}$	0.0321	0.0119	0.0080
τ	2.0484	0.2649	<0.0001

## Data Availability

The real dataset is freely available on the websites cited in the article. It also can be obtained by emailing the authors.

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
