# Peer review of "Modeling Overdispersed Dengue Data via Poisson Inverse Gaussian Regression Model: A Case Study in the City of Campo Grande, MS, Brazil"

_entropy, 2022, doi:10.3390/e24091256_

Round 1

Reviewer 1 Report

See attached file.

Author Response

We thank the reviewer for the detailed review of the manuscript and for the comments, suggestions, and criticisms.

Attached, we are sending the point-by-point replies to the comments done.

Best regards.

Reviewer 2 Report

The manuscript was generally well-written, and has potential to be considerably improved with some extra work. The authors need to better explain the originality and significance of the work.

Comments:

Can you say anything about the content of references [8], [13] and [10] in the Introduction?

The claim that the NBR is inadequate for long-tailed datasets in the Introduction should be substantiated with a reference. Same comment for the PIG having a larger range of skewness.

"in a real data set D" -> "to a real data set D"

The goals of the work should be more clearly stated in the Introduction. The novelty and significance of the work should be clearly explained.

Please correct "that there are available the measurements..." -> "that measurements of p explanatory variables are available".

"Probability function" -> "Probability mass function"

Section 3 begins with some interesting descriptions of the problem of dengue fever in Campo Grande, some of this should be brought forward to the Introduction.

To explain the significance of this work, can you reference any existing attempts to develop models for dengue surveillance in this area, and explain their weaknesses.

I'm not sure what you mean by "adjustment of a statistical model", please clarify.

Please provide a reference for variance inflating factors and explain to the reader briefly how they can be interpreted.

Please provide a brief description of GAMLSS and its capabilities for the reader.

Please explain how the Q-Q plot and worm plot should be interpreted.

The model design would likely be significantly improved by considering more flexible functions of the explanatory variables. This is possible in the GAMLSS package.

I would recommend presenting scatter plots of the explanatory variables against the dengue cases, with estimated relationships from the model.

Axis labels and Figure captions are not adequate for Figures 3-5. Please also correct "Figura" to "Figure".

Please detail how the 95% confidence bands were computed. They seem overly wide to me, especially in years when cases were low, indicating the model is possibly not explaining much of the variance. More flexible functions of covariates could improve this.

Forecasting capability is mentioned but should be explained and ideally tested quantitatively.

Author Response

(The authors gave the same response as above.)

Reviewer 3 Report

See attached report.

Author Response

(The authors gave the same response as above.)

Round 2

Reviewer 1 Report

Comments are in attached file.

Author Response

Please, follows attached the point-to-point answers to the raised questions.

Reviewer 3 Report

See attached. 

Author Response

(The authors gave the same response as above.)

Round 3

Reviewer 1 Report

I am satisfied with the comments added to the Discussion on the limitations of the study, in terms of not modelling the temporal correlation. 

Correction: In the second last sentence, "answer variable" should be "response variable".